# LFDe: A Lighter, Faster and More Data-Efficient Pre-training Framework for Event Extraction

Submission Id: 37

## ABSTRACT

Pre-training **E**vent **E**xtraction (EE) models on unlabeled data is an effective strategy that frees researchers from costly and labor-intensive data annotation. However, existing pre-training methodologies necessitate substantial computational resources, requiring high-performance hardware infrastructure and extensive training duration. In response to these challenges, this paper proposes a **L**ighter, **F**aster, and more **D**ata-**e**fficient pre-training framework for the EE task, named LFDe. Distinct from existing methods that strive to establish a comprehensive representation space during pre-training, our framework focuses on quickly familiarizing with the task format from a small amount of automatically constructed weak-label data. It comprises three stages: weak-label data construction, pre-training, and fine-tuning. Specifically, during the weak-label data construction stage, our framework first automatically designates pseudo triggers and arguments based on the characteristics of events in real datasets to form pre-training samples. In the processes of pre-training and fine-tuning, the framework reframes event extraction as the identification of words or phrases semantically closest to the prompt within the given sentence. This paper introduces a novel prompt-based sequence labeling model for EE to accommodate this reframing. By leveraging type-aware prompt features to augment original text embeddings, it enables the conventional sequence labeling model to extract events in data-scarce scenarios. Experiments on real-world datasets show that compared to similar models, our framework requires fewer pre-training instances (only about 0.04%), a shorter pre-training period (about 0.03%), and lower memory requirements (about 57.6%). Simultaneously, our framework significantly improves performance in various data scarcity scenarios.

**Relevance Statement**: EE aims to automatically extract event information from Internet text, providing a solid foundation for downstream tasks such as retrieval and recommendation systems.

## CCS CONCEPTS

• **Computing methodologies** → **Information extraction**.

## KEYWORDS

Event Extraction, Data-Efficient, Pre-training, Data Generation

## 1 INTRODUCTION

With the rapid advancement of network technology, there has been an exponential increase in the volume of digital textual data on the internet, providing substantial support for the development of information technology. However, the majority of these data consist of unstructured text, which is challenging for machines to understand and utilize. Therefore, researchers usually need to extract information from these texts first before they can leverage

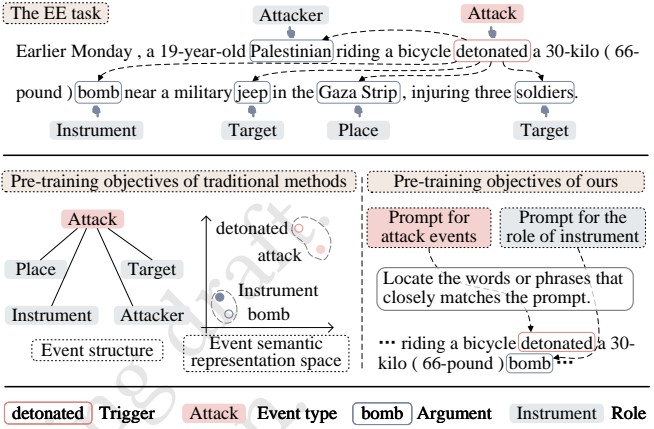

**Figure 1: The illustration of the event extraction task and an intuitive comparison between the pre-training objectives of traditional methods and ours.**

advanced algorithms and machines to analyze and research the digital data. Event extraction is a critical and challenging task within the field of information extraction. It aims to automatically detect event triggers and extract arguments within unstructured natural language text. Figure 1 intriguingly illustrates the goals of this task.

Existing EE methods [38, 45, 50] largely depend on manual annotations as supervised signals. Nevertheless, annotating data demands considerable manpower, economic investment, and significant time dedication, escalating the cost of model development. Large language models with notable comprehension and generation abilities, such as ChatGPT [1], can accomplish the EE task without the fine-tuning stage on manual annotations. However, recent researches [10, 48] have revealed much room for improvement in the performance of large models in this task. Pre-training EE models on large-scale unlabeled corpora provide an effective strategy to overcome annotation challenges, as is evidenced by both CLEVE [47] and UIE [31]. Unfortunately, the pre-training processes of CLEVE and UIE require high-performance hardware facilities, massive unlabeled corpora, and lengthy training periods. The pre-training of CLEVE involved 1.8 million articles and was computed using eight 2080Ti graphics cards, consuming a total of 72.5 hours. For UIE, the pre-training stage leveraged 8 NVIDIA A100 GPUs to train on 65 million samples with a batch size of 512 over 500,000 steps.

The lengthy training periods of CLEVE and UIE are attributed to the difficulty in quickly accomplishing their pre-training objectives. As shown in Figure 1, they are dedicated to modeling two types of features during their pre-training phases: event structure and

---
[1] https://openai.com/chatgpt

event semantic features. The former describes the components of an event. For example, the event with the type of "Attack" usually involves four kinds of event arguments: "Attacker", "Target", "Place", and "Instrument". The process of modeling event semantic features aims to fine-tune the representation space of **P**re-trained **L**anguage **M**odels (PLMs), which are used as backbones, to ensure that triggers or arguments of the same type are closely distributed. During the pre-training phase, CLEVE and UIE strive to construct a representation space covering the above-mentioned features. As a result, this leads to long training periods and high costs.

Although CLEVE and UIE have been successful in pre-training EE models, another question arises: how to reduce the cost and time overhead of the pre-training phase. Based on the above analysis, the answer to the question comes from two aspects: 1) Avoiding drastically adjusting the representation space of PLMs. 2) Simplifying the objective of the pre-training phase. However, implementing these strategies invites a couple of notable challenges: 1) The integration of event schemas into EE models without substantial adjustments to the representational space. 2) The simplification of training objectives to expedite the pre-training duration.

In this paper, we propose a lighter, faster and more effective pre-training framework for EE called LFDe. It addresses the first challenge above by transforming the task form of EE. Specifically, the crux of the EE task lies in detecting triggers and event arguments in sentences and then determining event types and argument roles. Therefore, this framework ingeniously transforms the task into a new form: locating the words or phrases that are semantically closest to type-aware prompts for specific event types or argument roles in the given sentences. Consequently, LFDe can learn event schema from prompts, eliminating the need to model event structure information from large-scale data. For the second challenge, instead of constructing a comprehensive event representation space, the pre-training objective of LFDe is to marginally adjust the existing representation space of the pre-trained language model to quickly adapt to new task formats. Our framework consists of three stages: weak label data construction, pre-training, and fine-tuning. In the first stage, the proposed framework automatically labels pseudo triggers via lexical annotation tools and locates pseudo arguments through **A**bstract **M**eaning **R**epresentation (AMR) from publicly available unlabeled text. Subsequently, it generates pseudo-labels for automatically labeled triggers and arguments through internet-based retrieval or **L**arge **L**anguage **M**odels (LLMs). Pre-training and fine-tuning stages are similar: LFDe generates type-aware prompts for each event type and argument role through pre-designed templates and works to locate triggers and arguments under the guidance of the prompts. To accommodate the novel form of EE introduced in this paper, we also propose a novel **P**rompt-based **S**equence **L**abeling **M**odel (P-SLM) to extract events. It treats locating triggers and arguments as sequence labeling and enhances token representations within each sentence using type-aware prompt features.

In summary, the contributions of this work are as follows:

- We propose a lighter, faster, and more data-efficient pre-training framework for EE. It utilizes 0.04% of the samples, 0.03% of the training time, and lower graphics card configurations of similar methods yet achieves superior performance.

- A novel prompt-based sequence labeling approach is proposed to adapt to the innovative event extraction task format brought about by the LFDe framework.
- We explore a novel pre-training objective for EE: subtly fine-tuning the representation space of PLMs to adapt to the task format of EE swiftly. It avoids drastically modifying the semantic representation space, which significantly reduces the pre-training period.
- This paper introduces an effective method for constructing weakly labeled data for EE. It yields weak label data that is effective for pre-training LFDe to familiarize with the EE task format proposed in this paper.

## 2 RELATED WORK

Earlier approaches [1, 12, 23, 34, 37] for the EE task manually design symbolic features based on lexical, syntactic and other rules and detect events and extract arguments via pattern matching. However, these patterns present a challenge in terms of generalization across distinct datasets. As a result, such methods excessively burden the user with complex feature engineering when applied for varied applications, thus necessitating a high level of expertise. The evolution of the deep learning technique has offered a promising solution to this predicament. Approaches based on neural networks [5, 39, 40, 43] automatically capture distributed features, enhancing models' transfer abilities while avoiding tedious feature engineering. In recent years, the increasing adoption of PLMs [7, 42] in the field of natural language processing has inspired EE methodologies to leverage various PLMs as backbones for model construction.

According to the backbone models and how the PLM is used, existing methods based on PLM can be classified into three categories: fine-tuning-based, prompt-based, and generative. "**Fine-tuning**"-based approaches [3, 9, 21, 25, 26] treat encoder-only PLMs as efficient encoders. They extract triggers and event arguments by feeding embeddings that output from PLMs into downstream networks designed for the EE task. In this paradigm, different downstream networks are constructed for various motives, yielding significant performance in scenarios suffused with data-abundant [50], few-shot [6] and zero-shot [53]. "**Prompt**"-based methods [15, 22, 28] are focused on eliciting potential knowledge of PLMs via inserting additional prompts into inputs. They bridge the gap between EE and pre-training phases by transforming the former into the format of pre-training tasks without altering the network structure. Benefiting from the prompt information and the knowledge PLMs learned during the pre-training phase, methods based on this framework could achieve impressive results on extremely scarce training data [13, 17, 33, 52]. "**Generative**" approaches [30, 31, 41] aim to directly output structured event information. They employ PLMs [20, 42] with encoder-to-decoder architectures to capture and produce events from texts, achieving commendable success in both data-scarce and data-abundant scenarios.

In addition to the above technical routes, recent years have seen the advent of other innovative schemes for the EE task. Unified information extraction approaches [38, 45] aim to uniformly model entity, relation, and event information, leveraging the intrinsic dependence among these diverse types of information. UIE [31] and OneIE [24] stand as exemplary representations of such technical

route. Kan et al. improved the performance of the fine-tuning paradigm by introducing prompts to input and generating auxiliary outputs [18]. This hybrid approach that integrates both fine-tuning and prompt paradigms demonstrated robust performance across various data scenarios. Following the emergence of LLMs, researchers explored the potential of utilizing them for event extraction [10, 48]. Studies indicate that LLM-based event extraction techniques exhibit commendable performance in the zero-shot scenario.

Annotating event data is expensive and laborious. Introducing external knowledge is an effective solution for extracting events in data-scarce scenarios. Liu et al. extended training datasets with FrameNet[2] [27], while Chen et al. used it and Freebase [2] to automatically annotate labels for given event types [4]. Based on WordNet[3], Wang et al. proposed a weakly supervised event detection method that automatically labels open domain data and then denoises the noise by adversarial training [46]. Yu et al. proposed a keyword clustering approach to generate event data for downstream training automatically [51].

## 3 BACKGROUND

This paper adopts the definition from **A**utomatic **C**ontent **E**xtraction[4] (ACE) evaluation: an event is a specific occurrence involving certain participants or states. The process of event extraction typically involves the following concepts:

- **Event type**: the category to which an event belongs.
- **Event trigger**: the word or phrase in the sentence that best reflects the occurrence of an event. It usually has the grammatical properties of a verb or noun.
- **Event argument**: the participants or attribute values (e.g. job-title, crime) of an event.
- **Argument role**: the role that an event argument plays within the event it participates in.

An event is composed of an event trigger and an uncertain number of event arguments. With the sentence in Figure 1 as the input, the aim of event extraction is four-fold:

- **Trigger detection**: identifying "*detonated*" as a trigger.
- **Event type classification (Event detection)**: determining the event type "*attack*" based on the trigger "*detonated*".
- **Event argument recognition**: identifying "*Palestinian*", "*bomb*", "*jeep*", "*Gaza Strip*", and "*soldiers*" in the sentence as arguments of the aforementioned event.
- **Role assignment (Argument classification)**: assigning roles to the event arguments discovered above.

## 4 METHOD

### 4.1 Task Formalization

Given a sentence $S = \{s_1, s_2, \cdots, s_{|S|}\}$, where $s_i$ represents the $i$-th token in the sentence, and $|S|$ is the length of the sentence, the goal of event extraction is to extract a set of events $\mathcal{E}$ under the guidance of the event schema $\mathcal{S}$. The event schema is artificially defined and outlines all the event types $T = \{t_1, t_2, \cdots, t_{|T|}\}$ and the argument roles $R = \{r_1, r_2, ..., r_{|R|}\}$ involved in each event

[2]https://framenet.icsi.berkeley.edu
[3]https://wordnet.princeton.edu
[4]http://projects.ldc.upenn.edu/ace/

Submission ID: 37. 2023-10-13 11:01. Page 3 of 1–11.

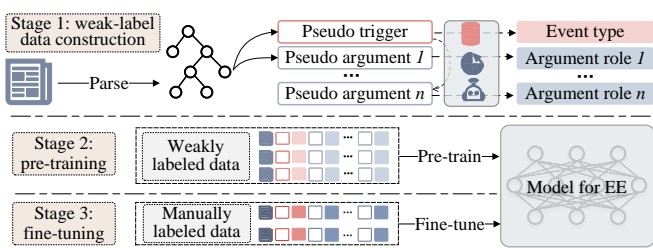

**Figure 2: The workflow of our proposed three-stage pre-training framework for event extraction. In the first stage, it generates pseudo triggers and pseudo arguments and automatically annotates pseudo-labels. In stages 2 and 3, the framework employs weakly labeled and manually labeled data to train the event extraction model, respectively.**

type, where $|T|$ is the number of event types, $R$ is the collection of argument roles included in the event with the $i$-th type, and $|R|$ is the length of $R$. The event set encompasses multiple components: $\mathcal{E} = \{E_1^{t_1}, E_2^{t_1}, \cdots, E_i^{t_j}, \cdots\}$, where $E_i^{t_j}$ denotes the $i$-th event of type $t_j$. Each event in the set $\mathcal{E}$ can be signified as: $E_i^{t_j} = \{\text{Trig}, \text{Arg}_1^{r_1}, \text{Arg}_2^{r_1}, \cdots, \text{Arg}_k^{r_l}, \cdots\}$, where Trig is the trigger, $\text{Arg}_k^{r_l}$ signifies the $k$-th argument with the role of $r_l$, and every element in event $E_i^{t_j}$ is a subset of the sentence $S$.

### 4.2 The Pre-training Framework

*4.2.1 Overall Design.* To avoid significantly altering the representation space of PLMs, LFDe integrates the event schema as prompts into the input. This effectively liberates the event extraction model from structuring event features. Concurrently, this framework simplifies capturing semantic characteristics of events by transforming the EE task into locating words or phrases that bear the strongest semantic alignments with the prompts within the given sentences. Specifically, LFDe generates type-aware (or role-aware) prompts for the predefined event types in the event schema and the argument roles involved in each event, forming the prompt sets $P_T$ and $P_A$. Defining $P_T$ and $P_A$ more specifically, they can be represented as $\{P^{t_1}, P^{t_2}, \cdots, P^{t_{|T|}}\}$ and $\{P^{a_1}, P^{a_2}, \cdots, P^{a_{|R|}}\}$ where $P^{t_i}$ refers to the type-aware prompt concerning the $i$-th event type, and $P^{a_i}$ pertains to the role-aware prompt regarding the $i$-th argument role. By incorporating the above type-aware (or role-aware) prompts into the original sentences, LFDe generates multiple type-specific inputs. Finally, the framework proposed in this paper extracts event triggers and arguments type-by-type from these inputs.

*4.2.2 Workflow.* As illustrated in Figure 2, the proposed framework trains the event extraction model in three stages. The first stage involves the process of automatically constructing weak label data. Specifically, LFDe parses publicly available text using **P**art-**O**f-**S**peech (POS) tagging and abstract meaning representation, selecting suitable words or phrases to serve as pseudo-triggers or pseudo-arguments. Subsequently, it employs LLMs to supplement the pseudo-arguments. Finally, the framework employs database querying, internet retrieval, and LLM-based question answering to yield pseudo-labels for the auto-annotated pseudo-triggers and

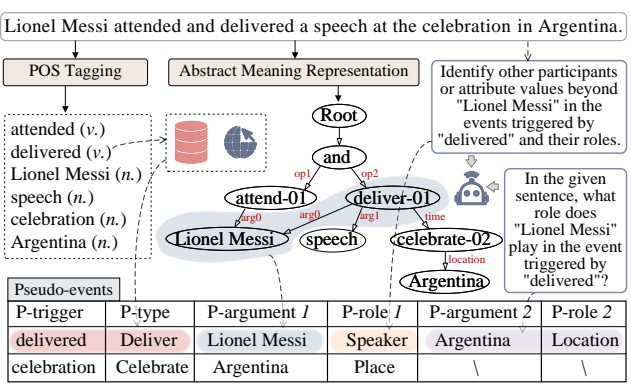

Figure 3: The overview of the weak-label data construction method proposed in this paper.

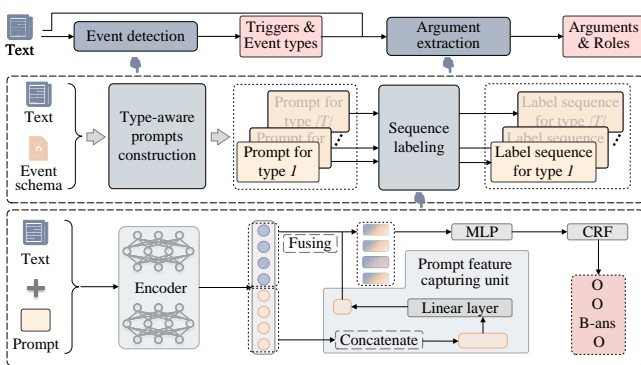

Figure 4: The schematic diagram of the prompt-based sequence labeling model for EE. It generates type-aware prompts and sequentially identifies triggers or arguments in the sentences under the guidance of these prompts.

pseudo-arguments, forming the weak-label dataset. In the second stage, our framework aims to utilize the automatically annotated weak-label data to pre-train the EE model. Finally, the framework fine-tunes the pre-trained model on manually labeled datasets.

## 4.3 Weak-label Data Construction

As shown in Figure 3, LFDe constructs weak-label data for pre-training by automatically annotating pseudo-events on unlabeled texts from public sources. Specifically, the framework collects news headlines from the Internet, which typically contain events, as the base corpus. It then utilizes POS tagging to filter pseudo triggers and leverages AMR to mine pseudo arguments, forming pseudo-events.

### 4.3.1 Pseudo-triggers Annotation.
As reported in previous studies [5], triggers are typically verbs or nouns. Therefore, LFDe applies part-of-speech tagging tools to label the tokens in public texts. Subsequently, it selects pseudo-triggers from the words or phrases that are tagged as verbs or nouns. The specific process is as follows:

*Verbs.* Initially, LFDe eliminates non-notional verbs unlikely to serve as triggers (such as "do", "can", "seem", etc.). Subsequently, it randomly selects verbs to serve as triggers of pseudo-event. It should be noted that, within actual datasets, some triggers are variations of event-type names. For instance, "married" could be the trigger for a "Life.Marry" event. Consequently, we utilize WordNet [35] to recover the verb's original form as the event type it triggered. In other scenarios, triggers are synonyms of event type names, such as the trigger "wed" within the "Life.Marry" event. Consequentially, LFDe selects pseudo-triggers' close synonyms from the WordNet at random to designate their types.

*Nouns.* In this situation, LFDe first discards nouns ending in specific suffixes (such as "-er/-or/-ist", "-ty/-ness", "-ing"), which commonly denote individuals, attributes, and present participles of verbs. Subsequently, it retrieves the evolution of each noun in the sentence from the online etymology dictionary[5]. If the evolution of the noun (such as "marriage") contains at least one verb ("marry"), then the proposed framework annotates the noun as a pseudo-trigger and randomly samples a verb as the event type.

[5]https://www.etymonline.com

### 4.3.2 Pseudo-arguments Annotation.
The annotation of pseudo-arguments is predicated on the existence of pseudo-triggers within the public text. Given a sentence $S$ and the previously labeled pseudo-trigger $Trig_p$, LFDe firstly abstracts the sentence to obtain an abstract meaning representation graph $G$ that represents its main semantics. Subsequently, as demonstrated in Figure 3, the framework locates the node corresponding to the pseudo-trigger word in $G$. It then marks the texts corresponding to the child nodes, which are connected to the pseudo-trigger node with "argx" edges, as pseudo-event arguments. Following this, LFDe identifies the role played by these pseudo-arguments in the pseudo-event through instruction learning by leveraging LLMs. Furthermore, our framework uses LLMs to supplement pseudo-arguments that are difficult to detect directly in $G$ and assign roles to them. The instructions used in this subsection are listed in Appendix A.

## 4.4 The Prompt-based Sequence Labeling Model

As depicted in Figure 4, the EE model proposed in this paper adopts a two-stage pipeline structure. With this structure, the model first detects events (identify triggers) and then extracts event arguments. The neural networks used in these two stages are identical, with the distinction lying in the prompts. In each stage, following the overall design presented in subsection 4.2.1, this model generates a type-aware prompt for every type $t_i$ in the event schema and subsequently extracts the triggers (or event arguments) from the sentence for each event type (or argument role). Specifically, the proposed EE model treats the extracting triggers (or arguments) as locating the span in a sentence semantically most relevant to the type-aware prompt through sequence labeling. The model's structure is illustrated at the bottom of Figure 4.

Drawing inspiration from the prompt-based studies [13, 14], we develop an enhanced type-aware prompt for each type (or role), incorporating type (or role) description, definition, examples, and typical triggers (or roles). Appendix A provides a detailed description of the construction of type-aware prompts. Given the sentence $S$ and the prompt $P^{t_i}$ of type $t_i$, the P-SLM initially combines them to form the type-aware input $I^{t_i} = \{s_1, s_2, \cdots, s_{|S|}, p_1^{t_i}, p_2^{t_i}, \cdots, p_{|P^{t_i}|}^{t_i}\}$,

where $p_j^{t_i}$ represents the $j$-th token in the prompt $P^{t_i}$, and $|P^{t_i}|$ denotes the total token count in $P^{t_i}$. It then feeds $I$ into a transformer-based encoder, producing the corresponding input embeddings $Emb^{t_i} = \{e_{s_1}^{t_i}, \cdots, e_{p_1}^{t_i}, \cdots\}$, where $e_{s_i}^{t_i}, e_{p_i}^{t_i} \in \mathbb{R}^h$, and $h$ is the hidden size. Subsequently, P-SLM obtains the prompt feature relevant to the type $t_i$ by applying the prompt feature capturing unit:

$$f_{Pt} = \text{ReLU}(\hat{e}_{Pt} W_P + b_P),$$
$$\hat{e}_{Pt} = [e_{p_1}^{t_i}; \cdots; e_{p_{|P^{t_i}|}}^{t_i}], \tag{1}$$

where $W_P \in \mathbb{R}^{|P^{t_i}| * h \times h}$, $b_P \in \mathbb{R}^h$, and $\text{ReLU}(\cdot)$ indicates the **Re**ctified **L**inear **U**int (ReLU) activation [11]. Following this, our model merges $f_{Pt}$ with the embeddings of every token in the sentence, formulating an advanced sentence embedding sequence denoted as $E_S^t$:

$$E_S^t = \{[s_1^{t_i}; f_{Pt}], [s_2^{t_i}; f_{Pt}], \cdots, [s_{|S|}^{t_i}; f_{Pt}]\} \tag{2}$$

Finally, the model applies the **M**ulti-**L**ayer **P**erceptron (MLP) and **C**onditional **R**andom **F**ield (CRF) algorithms to locate the span in $E_S$ that is semantically most relevant to the prompt.

## 4.5 Pre-training and Fine-tuning

In the pre-training phase, LFDe utilizes the weak label data generated in section 4.3 as the training set, while the fine-tuning phase employs manually annotated data to refine the model. The optimization objectives for both the pre-training and fine-tuning stages are identical, striving to minimize the negative log-likelihood loss:

$$\mathcal{L} = -\sum_{j=1}^{|D|} \sum_{i=1}^{|T|} \log\left(\frac{\text{Socre}_{\text{Real}}^i(S^j)}{\text{Socre}_{\text{Total}}^i(S^j)}\right), \tag{3}$$

where $|D|$ indicates the size of the (pre-) training set. "Score$(S^j)$" denotes the intrinsic score assigned by the CRF algorithm to a specific label sequence of the $j$-th sentence within the (pre-) training set. The ground truth for the $i$-th type is indicated by $\text{Socre}_{\text{Real}}^i(\cdot)$, and $\text{Socre}_{\text{Total}}^i(\cdot)$ signifies the cumulative score of all possible label sequences for the $j$-th data on the $i$-th type.

The weak-label data distribution differs significantly from the manually annotated data used in the fine-tuning stage. To alleviate the problem of potentially catastrophic forgetting caused by migration between two datasets, we strive to improve the model's generalization during the pre-training stage. We use **S**harpness-**A**ware **M**inimization (SAM) [19] in the pre-training stage, which seeks the minimum in the flat loss hyperplane and can effectively improve the generalization ability of the model.

## 5 EXPERIMENT

### 5.1 Experimental Settings

*5.1.1 Implementation Details.* News headlines utilized in this study are sourced from the "**G**lobal **D**atabase of **E**vents, **L**anguage, and **T**one (GDELT)" project[6]. Leveraging this resource, our framework autonomously generates 30,000 weak-label instances, of which 1,000 are used for evaluation during the pre-training stage while the remainder is used for pre-training. We utilize NLTK[7] to annotate part-of-speech tags and employ the same approach as CLEVE [49]

[6]https://www.gdeltproject.org/
[7]https://www.nltk.org/

**Table 1: Hyperparameters for the pre-training stage.**

| Hyperparameter | Backbone | Value |
|---|---|---|
| Learning rate | base & large | 1e-5 |
| Dropout | base & large | 0.5 |
| Step | base & large | 1000 |
| Batch size | base | 64 |
| | large | 32 |

**Table 2: Hyperparameters for the fine-tuning stage.**

| Hyperparameter | Scenario | Value |
|---|---|---|
| Learning rate | All | 1e-5 |
| Dropout | All | 0.5 |
| Epoch | All | 30 |
| Batch size | Data-abundant | 32 |
| | Low-resource (1%, 2%, 3%) | 8 |
| | Low-resource (5%, 10%, 20%) | 16 |
| | Low-resource (30%, 50%) | 32 |
| | Zero-shot | 32 |

to obtain AMR graphs. During the fine-tuning phase, performance is assessed based on the highest-scoring checkpoint on the development set. Across both the pre-training and fine-tuning stages, the training of LFDe is conducted on a single NVIDIA RTX A6000 GPU with 48G of video memory, employing RoBERTa-large [29] as the fundamental structure. The version of ChatGPT employed in this paper is "GPT-3.5-Turbo-16k-0613". The optimal parameters for LFDe during the pre-training and fine-tuning stages, determined through comprehensive grid searching with the F1 score as the pivotal criteria, are illustrated respectively in Table 1 and Table 2.

*5.1.2 Datasets and Evaluation.* Following previous studies [13, 24, 38], we conducted experiments on two datasets: **ACE-2005** [8] and **ERE** [44]. The former, ACE-2005, is the most widely used EE dataset featuring 33 types, whereas ERE is a more expansive dataset, encompassing 38 types. Notably, two variants of ACE2005 have emerged in recent years, namely ACE-05E and ACE-05E$^+$ [24]. The former filters out events with triggers composed of multiple tokens, while the latter does not. We evaluate our framework on the more comprehensive version **ACE-05E$^+$**. For both above datasets, we employ the data pre-processing procedure and data splits provided by previous studies [13, 24]. For the sake of fairness, we follow the data settings most commonly used in the event extraction field in recent years [13, 24]. Specifically, for the data-abundant scenario, we employ the same data split as that utilized in baseline approaches. In the low-resource regime, we form the training set using 1%, 2%, 3%, 5%, 10%, 20%, 30% and 50% of training samples identical to DEGREE [13] while maintaining the validation and test sets from the data-abundant scenario. In the zero-shot learning situation, we follow the method outlined in previous research [32], training our model on the ten most frequently occurring types in the ACE-2005 dataset and subsequently evaluating it on the remaining types.

We apply the same evaluation metrics as prior works [13, 14]. A trigger is accurately predicted if its type and span match the actual

Table 3: F1 scores in low-resource regime on ACE-2005 and ERE.

| | | | | | | | | | | | | | | | | |
|---|---|---|---|---|---|---|---|---|---|---|---|---|---|---|---|---|
| **Event Detection / Trigger Classification F1-Score (%)** | | | | | | | | | | | | | | | | |
| Model | ACE-2005 | | | | | | | | ERE | | | | | | | |
| | 1% | 2% | 3% | 5% | 10% | 20% | 30% | 50% | 1% | 2% | 3% | 5% | 10% | 20% | 30% | 50% |
| EEQA† | 18.9 | 29.4 | 47.4 | 52.6 | 54.2 | 61.0 | 63.4 | 66.1 | - | - | - | - | - | - | - | - |
| OneIE | 39.0 | 41.1† | 52.5 | 60.6 | 58.1 | 66.5 | 66.4 | 69.1† | 11.0 | 30.4† | 36.9 | 46.7 | 48.8 | 51.8 | 53.5 | 54.1† |
| Text2Event | 15.7 | 32.3† | 38.4 | 43.9 | 46.3 | 56.5 | 62.0 | 66.2† | 6.3 | 16.7† | 25.6 | 33.5 | 42.4 | 46.7 | 50.1 | 52.9† |
| CLEVE† | 32.4 | 36.7 | 43.8 | 46.0 | 53.2 | 58.2 | 62.1 | 65.3 | - | - | - | - | - | - | - | - |
| UIE† | 39.6 | 48.4 | 52.8 | 57.0 | 61.6 | 65.6 | 65.5 | 66.7 | - | - | - | - | - | - | - | - |
| DEGREE | 49.5 | 54.5† | **63.5** | 62.3 | **68.5** | 67.6 | 66.9 | 67.8† | 27.9 | 39.3† | 45.5 | 47.0 | 53.0 | 51.9 | 53.5 | 53.9† |
| LFDe | **54.4** | **59.2** | 57.7 | **63.4** | 65.4 | 68.2 | 68.8 | **70.1** | **35.9** | **42.5** | **46.6** | **49.7** | **53.8** | **52.7** | **54.3** | **54.7** |
| -*w/o* pre-training | 38.2 | 45.4 | 53.5 | 61.3 | 65.2 | 64.5 | 66.1 | 69.7 | 17.9 | 28.8 | 40.6 | 47.1 | 50.1 | 51.3 | 51.7 | 52.9 |
| **Argument Extraction / Argument Classification F1-Score (%)** | | | | | | | | | | | | | | | | |
| Model | ACE-2005 | | | | | | | | ERE | | | | | | | |
| | 1% | 2% | 3% | 5% | 10% | 20% | 30% | 50% | 1% | 2% | 3% | 5% | 10% | 20% | 30% | 50% |
| EEQA† | 5.2 | 9.1 | 13.4 | 22.6 | 26.2 | 39.7 | 46.7 | 50.6 | - | - | - | - | - | - | - | - |
| OneIE | 10.4 | 15.3† | 20.6 | 29.7 | 35.5 | 46.7 | 48.0 | 52.2† | 2.6 | 16.5† | 20.3 | 29.7 | 35.1 | 40.7 | 43.0 | 45.4† |
| Text2Event | 5.7 | 8.9† | 16.5 | 21.3 | 26.4 | 35.2 | 42.1 | 42.9† | 2.3 | 9.7† | 15.2 | 23.6 | 28.7 | 35.7 | 38.7 | 44.8† |
| CLEVE† | 12.8 | 15.1 | 21.7 | 25.2 | 31.3 | 37.9 | 42.1 | 46.8 | - | - | - | - | - | - | - | - |
| UIE† | 16.9 | 18.1 | 25.9 | 27.9 | 35.0 | 41.5 | 43.6 | 47.7 | - | - | - | - | - | - | - | - |
| DEGREE | 18.7 | 25.8† | 34.0 | 35.7 | 43.6 | 48.9 | 51.2 | 51.1† | 14.5 | 23.2† | 28.9 | 33.4 | 41.7 | 42.9 | 45.5 | 47.4† |
| AMPERE† | 21.4 | 28.6 | 33.7 | 37.2 | 43.8 | 48.2 | 51.4 | 52.4 | 16.1 | 24.8 | 30.4 | 34.1 | 40.5 | 42.2 | **46.2** | 47.7 |
| LFDe | **27.4** | **31.7** | **34.3** | **38.4** | **44.8** | **49.6** | **51.9** | **53.6** | **17.3** | **26.3** | **32.4** | **35.4** | **42.1** | **44.0** | 45.6 | **48.3** |
| -*w/o* pre-training | 10.6 | 17.4 | 26.8 | 28.4 | 38.2 | 46.9 | 49.9 | 50.8 | 11.6 | 19.3 | 25.3 | 29.7 | 34.2 | 36.5 | 43.5 | 45.4 |

value. An argument is correct when its event type, the trigger span, and the role match the gold one.

### 5.1.3 Baselines.

*5.1.3 Baselines.* We select seven advanced approaches that encompass four common techniques for EE using PLM, serving as baselines in the low-resource and data-abundant scenarios:

- EEQA [9] is a "fine-tuning"-based method, re-formulating the EE task as question answering.
- OneIE [24] also employs the fine-tuning paradigm. It uniformly models entities, relations, and events and benefits from their inter-dependencies.
- Text2Event [30] is representative of generative EE methods, which directly produce structured events.
- CLEVE [47] is a pre-training-based EE methodology leveraging contrastive learning techniques to learn semantic and structural information about events from unsupervised data.
- UIE [31] is a pre-training-based, unified framework that learns various extraction tasks from unsupervised data and directly yields structural information.
- DEGREE [13] is a method based on the "prompt" paradigm, which transforms EE into the pre-training task of the PLM. This paper selects the DEGREE with a joint structure as the baseline, as it generally performs better in data-scare scenarios compared to its pipeline-structured version.
- AMPERE [14] is a prompt-based model that incorporates AMR in prompts. As AMPERE is a data-efficient model designed exclusively for argument extraction, we employed LFDe as the event detection method for this baseline.

The symbol "†" indicates the results obtained by rerunning publicly available codes. Since most of the above baseline methods confront difficulties in extracting events with types that have never been seen, we also select "Transfer" [16], "ILP" [53], and "TE/QA" [32] as baselines in the zero-shot scenario. These methods aim to develop EE models with transfer ability, performing well on newly emerging event types. Additionally, we introduce an event extraction methodology based on ChatGPT, named "ChatGPT-EE", as a strong baseline for LFDe in the zero-shot scenario. The details of "ChatGPT-EE" are elaborated in Appendix C. LFDe-base refers to the use of "RoBERTa-base" as the backbone.

## 5.2 Main Results

*5.2.1 Low-resource.* Table 3 shows the F1 scores of the LFDe and baseline methods with different proportions of training instances on ACE-2005 and ERE. "-*w/o* pre-training" denotes directly training the P-SLM introduced in subsection 4.4 on the manually labeled datasets without the pre-training phase. The best scores are marked in bold, and the second-highest scores are underlined. It can be observed that our LFDe achieves the highest F1 scores across most data configurations for both event detection and argument extraction. In the remaining three cases, it demonstrates a commendably comparative performance. In particular, compared with the current state-of-the-art data-efficient methods DEGREE and AMPERE, which take a larger PLM "BART-large" as their backbones, our method has a significant improvement in most settings on both ACE-2005 and ERE. In contrast to UIE and CLEVE, which are also trained on unlabeled data, LFDe substantially improved F1 scores

**Table 4: Results (%) on ACE-2005 in the zero-shot scenario.**

| Model | Trigger | | | Argument | | |
|---|---|---|---|---|---|---|
| | P | R | F1 | P | R | F1 |
| Transfer | 75.5 | 36.3 | 49.1 | 16.1 | 15.6 | 15.8 |
| ILP | 54.1 | 53.1 | 53.6 | 4.6 | 10.0 | 6.3 |
| TE/QA | - | - | 41.7 | - | - | 16.8 |
| ChatGPT-EE[†] | 70.7 | 41.7 | 52.5 | 14.4 | 14.3 | 14.3 |
| DEGREE[†] | 52.4 | 53.7 | 53.1 | 45.1 | 15.3 | 22.8 |
| LFDe | 57.7 | 60.1 | **58.8** | 27.2 | 32.2 | **29.6** |
| -*w/o* pre-training | 54.2 | 56.8 | 55.5 | 21.1 | 27.4 | 23.8 |
| LFDe-base | 49.2 | 53.7 | 51.3 | 22.1 | 24.6 | 23.2 |

**Table 5: F1 scores (%) in the data-abundant scenario.**

| Model | ACE-2005 | | ERE | |
|---|---|---|---|---|
| | Trigger | Argument | Trigger | Argument |
| EEQA[†] | 70.9 | 52.4 | - | - |
| OneIE | **72.8** | 54.8 | 57.0 | 46.5 |
| Text2Event | 71.8 | 54.9 | **59.4** | 48.3 |
| CLEVE[†] | 71.2 | 54.5 | - | - |
| UIE[†] | 71.4 | 55.2 | - | - |
| DEGREE | 70.9 | **56.3** | 57.1 | 49.6 |
| AMPERE | - | 55.1 | - | 50.7 |
| LFDe | 71.8 | 55.1 | 58.0 | **51.4** |
| -*w/o* pre-training | 69.8 | 53.2 | 57.8 | 48.6 |

in all data settings. It shows that the introduced lighter pre-training framework is more data-efficient than conventional pre-training methodologies. The above observations demonstrate the effectiveness of the LFDe proposed in this paper.

We can draw three important conclusions by comparing "-*w/o* pre-training" with the other results. 1) The performance degrades noticeably without the pre-training phase, especially in settings with extreme data sparsity. It indicates that pre-training on a small amount of weakly labeled data to familiarize the model with the task format can effectively enhance its performance when manual annotations are sparse. 2) "-*w/o* pre-training" significantly outperforms OneIE and EEQA in the case of relatively sparse training data and performs close to them in the situation of more data. The comparisons suggest that the P-SLM enhanced with type-aware prompts is more competent in handling scenarios with sparse data. 3) The substantial outperformance of "-*w/o* pre-training" relative to Text2Event shows the advantages of the P-SLM over the traditional "generative" paradigm in low-resource settings.

*5.2.2 Zero-shot.* Since the zero-shot setting in the DEGREE paper is different from the one followed in this paper, the results of DEGREE in this experiment are obtained by rerunning their code. As shown in Table 4, LFDe significantly outperforms the strongest baseline method by 5.2% in event detection and 6.8% in argument extraction. Compared to the ChatGPT-EE, which is highly anticipated in the zero-shot setting, LFDe also achieves remarkable advancements in the classification performance of triggers and arguments. These improvements demonstrate the effectiveness of our framework in the zero-shot scenario. In addition, without the pre-training phase, our model also remarkably outperforms all baseline methods. This suggests that the prompt-based sequence labeling EE model proposed in this paper has a solid ability to transfer to new event types. The introduction of the pre-training phase causes a noticeable increase in the F1 score, underscoring the advantage of adapting the task format on unlabeled data for effective trigger and argument extraction in the zero-shot scenario.

*5.2.3 Data-abundant.* Although LFDe is designed for data-scarce scenarios, we conduct the experiment in the data-abundant scenario for controlled comparisons. Table 5 shows the results evaluated on the complete ACE-2005 and ERE. In terms of event detection, LFDe outperforms most baseline methods on two datasets, achieving the

second-highest F1 scores. As for argument extraction, our framework ranks third in F1 values on ACE-2005, while it takes the top position on ERE. Compared to other pre-training-based methods, our framework demonstrates competitive performance on ACE2005. It achieves the highest score in event detection and only lags slightly behind UIE in argument extraction. In comparison with the current state-of-the-art data-efficient methods, DEGREE and AMPERE, our framework outperforms in most cases. Generally, the method proposed in this paper achieves competitive results, demonstrating the effectiveness of LFDe in the data-abundant scenario.

Compared to the data-scarce scenarios, LFDe fails to achieve significant performance gains on ACE-2005 under the fully supervised setting. We attribute this critical factor: OneIE and UIE benefit better from the inherent dependencies between entities, relationships and events in sufficient training data. The comparison with "-*w/o* pre-training" illustrates that adapting the form of the EE task during the pre-training process can still help improve the performance of the P-SLM in the data-abundant scenario.

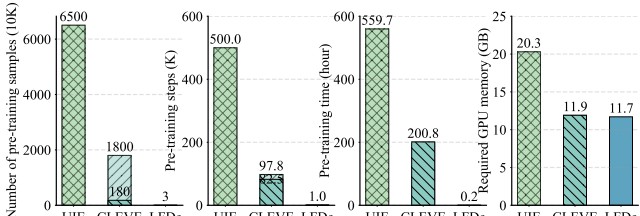

**Figure 5: Statistical analysis of overheads on various items during the pre-training phase.**

*5.2.4 Analysis of Pre-training Overheads.* Figure 5 displays the pre-training overheads of LFDe compared with two other pre-training-based EE methods. Within this visual, ▨ signifies the conservative value of CLEVE on the corresponding statistical item, while ▨ denotes its estimated value. Appendix D details the statistical methodologies employed within each item. The figure decisively underlines that our framework considerably minimizes pre-training overheads compared to both UIE and CLEVE. The observed advantages in the count of pre-training samples, steps involved in the pre-training phase, and reduced time expenditure illustrate that LFDe is a faster

pre-training framework for EE. Further, the advantage associated with GPU memory consumption suggests that our framework is lighter than existing approaches.

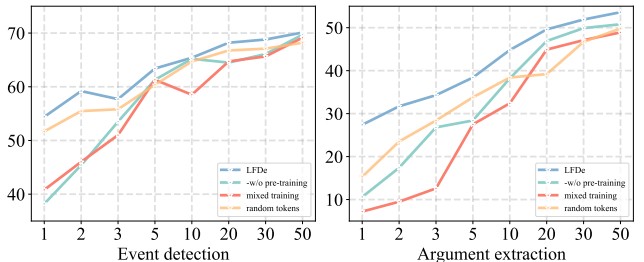

**Figure 6: Trends in results under different training strategies.**

## 5.3 Ablation Study

*5.3.1 The Effect of Pre-training.* Figure 6 visualizes the performances achieved by applying different training strategies on ACE-2005 in low-data settings. The horizontal axis represents the percentage of training data utilized, while the vertical axis measures the F1 scores. "Random tokens" denotes replacing public news headlines with random text strings. These strings comprise random words or phrases derived from the news domain, containing at least one verbal or nominal word (or phrase) serving as a trigger or an argument. "Mixed training" indicates mixing weak-label samples with manually labeled data as the training set. The performance comparison between LFDe, "- w/o pre-training" and "mixed training" reveals the potential advantage of pre-training EE models using automatically generated weakly labeled data, as it significantly enhances P-SLM's performance. Furthermore, the comparison involving "random tokens", "- w/o pre-training" and "mixed training" demonstrates that despite the absence of semantic knowledge within the pre-training corpus, providing a platform for the EE model to acquaint itself with the downstream task format can augment the model's performance in scenarios where data is extremely scarce.

**Table 6: Results (%) on ACE-2005 without finetuning.**

| Model | Trigger | | | Argument | | |
|---|---|---|---|---|---|---|
| | P | R | F1 | P | R | F1 |
| CLEVE | 0.8 | 6.1 | 1.4 | 0.4 | 0.7 | 0.5 |
| UIE | 0.3 | 3.4 | 0.5 | 0.0 | 0.4 | 0.0 |
| EEQA (1%) | 34.0 | 13.1 | 18.9 | 11.4 | 3.4 | 5.2 |
| Text2Event (1%) | - | - | 15.7 | - | - | 5.7 |
| ChatGPT-EE | 53.7 | 24.8 | 34.0 | 15.1 | 9.2 | **11.5** |
| LFDe | 31.4 | 38.9 | **34.7** | 8.2 | 6.2 | 7.0 |
| - random tokens | 33.8 | 16.7 | 22.3 | 2.4 | 0.8 | 1.2 |

*5.3.2 The Effect of Pre-learning Task Formats.* To visualize the benefits of pre-learning task formats, we directly evaluate our model on the test set used in the low-resource scenario after pre-training and present the results in Table 6. EEQA(1%) and Text2Event(1%) denote fine-tuning EEQA and Text2Event on 1% of the training data. Without fine-tuning, LFDe achieves remarkable F1 scores of 34.7% and 7.0% in event detection and argument extraction, respectively. In addition, it significantly outperforms Text2Event(1%) and EEQA(1%) when pre-training P-SLM with samples consisting of random words. With affordable pre-training, our approach even surpasses ChatGPT-EE in event detection. The above observations demonstrate that pre-learning the task format can effectively contribute to the EE task. Comparisons with UIE and CLEVE show that the EE framework proposed in this paper is more data-efficient. LFDe's substantial improvement over "random tokens" implies that enhancing the quality of the weak label data can notably bolster the task format knowledge obtained during the pre-training phase.

**Table 7: F1 scores (%) on ACE-2005 with different prompts.**

| Model | 1% | 3% | 5% | 20% | 50% |
|---|---|---|---|---|---|
| | | | Trigger | | |
| LFDe | 54.4 | **57.7** | **63.4** | 67.3 | **70.1** |
| - *w/o* type description | **54.5** | 55.3 | 60.8 | 66.4 | 67.3 |
| - *w/o* definition | 53.9 | 54.6 | 59.1 | 67.6 | 66.7 |
| - *w/o* examples | 54.1 | 55.9 | 60.8 | 65.6 | 67.8 |
| - *w/o* typical triggers | 50.3 | 53.4 | 56.6 | 64.9 | 67.6 |
| | | | Argument | | |
| LFDe | **27.4** | **34.3** | **38.4** | **49.6** | **51.9** |
| - *w/o* role description | 26.1 | 32.9 | 36.4 | 48.8 | 50.7 |
| - *w/o* definition | 26.2 | 33.5 | 37.3 | 48.2 | 51.5 |
| - *w/o* examples | 25.8 | 31.2 | 35.1 | 47.5 | 50.4 |
| - *w/o* typical arguments | 26.1 | 32.5 | 35.7 | 47.5 | 50.3 |

*5.3.3 The Effect of Prompts' Components.* We study the effects of the four components of type-aware prompts by individually removing each of them. It can be observed that the removal of each prompt component causes performance degradation in almost every low-resource setting. In event detection, "typical triggers" exert the most significant influence on performance. Meanwhile, the "examples" part, besides the "typical arguments", also substantially impacts the model's performance in argument extraction. These observations reciprocally corroborate with the In-context learning theory [36]. Leveraging this theory, we derive the following conclusions: 1) Listing typical words in prompts aids the model in locating similar triggers or arguments. 2) Examples of argument extraction can provide additional knowledge to help the model clarify the relationship between event elements and event types.

## 6 CONCLUSION

We propose a lighter, faster, and more data-efficient pre-training framework and a prompt-based sequence labeling model for event extraction in data-scarce scenarios. The framework formalizes both event detection and argument extraction as locating spans in sentences under the guidance of prompts and familiarizes EE models with the above task formats using public unlabeled corpora. Our framework proposed in this paper first pre-learnings the task format on automatically generated weak-label data and then fine-tunes the pre-trained model on manually labeled datasets. Experiments on real-world data demonstrate the effectiveness of our framework.

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

## A  INSTRUCTIONS USED IN SUBSECTION 4.3.2

**Table 8: Instructions for pseudo-arguments annotation.**

| Purpose | Instruction |
|---|---|
| Roles labeling | Given the sentence "[**sentence**]", what role does "[**pseudo argument**]" play in the "[**event type**]" event triggered by "[**pseudo trigger**]"? |
| pseudo-arguments supplementing | Given the sentence "[**sentence**]", are there any other participants in the event of type "[**event type**]", triggered by "[**pseudo trigger**]", other than "[**pseudo argument 1**]", "[**pseudo argument 2**]", ..., and "[**pseudo argument** $m$**]"? If so, output them sequentially in the format "Participant Role: Participant". If not, please output "No". Output the answer directly without any explanations. "[**Examples**]". |

The instruction templates used in subsection 4.3.2 are presented in Table 8. In practical application, our framework replaces the bold text in the templates with actual contents such as news headlines, pseudo-triggers, event types, and pseudo-arguments. "[**Examples**]" consists of several input-output pairs, in which the input includes a given news headline, event type, trigger, and partial event arguments. On the other hand, the output is the "participant role: participant" pair that needs to be completed in this step.

## B  TYPE-AWARE PROMPTS CONSTRUCTION

**Table 9: Components of type-aware prompts.**

| Stage | Component | Content |
|---|---|---|
| Trigger | Type description | The sentence "[**sentence**]" contains an event about "[**event type**]". |
| | Definition | This type of event is defined as "[**type definition**]". |
| | Examples | The following is an example of this event: "[**trigger examples**]". |
| | Typical triggers | Typical triggers for this type of event are: "[**typical triggers**]". |
| Argument | Role description | In the sentence "[**sentence**]", an event of type "[**event type**]" is triggered by "[**trigger**]", encompassing the argument of "[**argument role**]". |
| | Definition | A "[**argument role**]" argument is defined as "[**role definition**]". |
| | Examples | The following is an example of this argument role: "[**role examples**]". |
| | Typical arguments | Typical arguments for this role are: "[**typical arguments**]". |

The type-aware prompts comprise four elements: type description, definition, examples, and typical words. Table 9 illustrates the templates for these four components during the trigger extraction and argument extraction stages. Any bold text within the table should be replaced with real values during practical implementation. "[**type definition**]" and "[**role definition**]" are type (or role) definitions from the ACE2005 English corpus annotation guidelines. "[**trigger examples**]" and "[**role examples**]" represent the examples provided in the guidelines. We continue to use typical triggers used in the DEGREE research as "[**typical triggers**]". We meticulously selected multiple typical event arguments for each role from the ACE2005 training set severing as "[**typical arguments**]".

## C  THE CHATGPT-BASED APPROACH FOR EVENT EXTRACTION

---
**Algorithm 1** Pipeline-based Event Extraction

---
**Input:** Sentence $S$, Prompt library $\mathcal{P}$
**Output:** Structural events $\mathcal{E}$
1: Initiate $\mathcal{E} \leftarrow \varnothing$
2: Get the trigger prompt $P_{\text{trig}}$ from $\mathcal{P}$
3: $Req \leftarrow P_{\text{trig}}$
4: Get response $Resp$ of request $Req$ from ChatGPT
5: Parse triggers $Trigs$ from $Resp$
6: **for** $trig$ in $Trigs$ **do**
7:     Parse the event type $t$ of $trig$ from $Resp$
8:     Retrieve the prompt $p_t$ from $\mathcal{P}$ of event type $t$
9:     $Req \leftarrow Req \bigoplus Resp \bigoplus p_t$
10:     Get response $Resp$ of request $Req$ from LLMs
11:     Parse arguments $Args$ from $Resp$
12:     Append $(trig, Args)$ to $\mathcal{E}$
13: **end for**

---

Our ChatGPT-based EE method employs a pipeline structure, which detects events first and then extracts event arguments based on the predicted triggers and event types. The workflow of ChatGPT-EE is shown in Algorithm 1. We employ the In-context learning technique to guide ChatGPT in producing dictionary-format results of the event detection phase and argument extraction phase, facilitating the data post-processing program to parse event information from the response of ChatGPT. The instruction for event detection is "*Extract event triggers with the type [**event type**] from the sentence [**sentence**]. Use the following format to output all the event information in the text sequentially: TRIG:{'event type': 'trigger'}. If the sentence does not contain any events, just output "No event". Output the answer directly without any explanation.*". The instruction used in the argument extraction stage is "*Given sentence [**sentence**], it encompasses an event of type [**event type**], which is triggered by [**trigger**]. This type of event comprises three argument roles: [**argument role 1**], [**argument role 2**], ... . Use the following format to output all the argument in the text sequentially: ARG:{'role 1': ['argument 1_1', 'argument 1_2', ...], 'role 2': ['argument 2_1', 'argument 2_2', ...], ...}. If the event does not contain any argument, just output "No argument". Output the answer directly without any explanation.*"

# D STATISTICAL ANALYSIS OF PRE-TRAINING OVERHEADS

## D.1 Number of Pre-training Samples

We obtained the quantity of pre-training samples from the original paper of UIE [31]. The original CLEVE paper [47] mentioned that they utilized 1,800,000 articles from The New York Times serving as the training corpus. Assuming that each article in this corpus contains an average of 1 sentence, we can consider 1,800,000 as a conservative value of pre-training samples for CLEVE. We randomly sampled ten articles from this corpus, yielding an average sentence count of 10.4. Thus, we can hypothesize that the NYT corpus's sentence count is approximately 18,000,000, which we take as the estimated value.

## D.2 Pre-training Steps

The values for the pre-training steps of UIE are drawn from its paper. In the case of CLEVE, the pre-training steps comprise two parts: event semantic pre-training and event structure pre-training. The original paper for CLEVE reported the number of training steps in the event structure pre-training phase as 82,500, but the number of steps in event semantic pre-training was not reported. We took 82,500 as a conservative value and roughly estimated the number of training steps for event structure pre-training to be 15,300, based on the ratio of training time costs between the event semantic pre-training stage and the event structure pre-training stage. Although there are inevitable errors between the estimated and actual values, this provides us with a reference value.

## D.3 Pre-training Time

We independently trained the UIE and CLEVE models for 1000 steps on an A6000 graphics card using recommended parameter settings and recorded the time expenditure of the above process. Subsequently, we estimated their time costs according to the pre-training steps mentioned in their papers.

## D.4 Required GPU Memory

We standardized the input size for UIE, CLEVE, and LFDe by utilizing the same batch size and maximum input length and trained them on a single A6000 graphics card. During training, we record the peak of GPU memory utilization for comparison.

Received 20 February 2007; revised 12 March 2009; accepted 5 June 2009

