# OpenReview forum: "LFDe: A Lighter, Faster and More Data-Efficient Pre-training Framework for Event Extraction"
_ACM.org/TheWebConf/2024/Conference — TheWebConf24_

### Official Review · Reviewer_bMRv · 2023-11-06

**Novelty:** 5
**Technical Quality:** 5

**Review:**

The paper proposes LFDe which is a new method for Event Extraction (EE). LFDe presents a more effective and efficient pre-training for EE by learning event schema from prompts which eliminates the need to learn the event structure information from large-scale data. A weak-label data is automatically constructed for the pre-training phase by first  locating pseudo-triggers with POS tagging and pseudo-arguments with AMR, and then pseudo-roles are inferred through instruction learning by leveraging LLMs. A prompt-based sequence labeling model (P-SLM) is introduced for both event detection and argument extraction phases using type-aware prompts and role-aware prompts, respectively. The final layer of P-SLM is a CRF that locates spans of triggers of event types and roles of arguments.

Pros

S1. The authors proposed a more effective and efficient pre-training of EE by introducing event prompts for both event detection and argument extraction phases that familiarize the EE model with the task format using public unlabeled corpora.

S2. The weak-label data that is used in the pre-training phase is constructed automatically by first  locating pseudo-triggers with POS tagging and pseudo-arguments with AMR, and then pseudo-roles are inferred through instruction learning by leveraging LLMs.

S3. The authors compared their method to many existing baselines, and showed improvements in the evaluation metrics for both event detection and argument extraction. The ablation study is also interesting as it shows the effect of pre-training, the effect of pre-training task formats, and the effect of prompts’ components.

Cons

W1. The authors mentioned that Global Database of Events, Language, and Tone (GDELT) is used for pre-training. The choice of this particular data should be justified. In addition, for fair comparison, similar pre-training datasets should be used with the baselines.

W2. The event schema which is integrated as prompts into the input is an important part of the model, but it is not clear how this event schema is constructed.

W3. For the fully fine-tuning case (Table 5), it is not clear why the event schema is not as helpful as in the low-resource scenarios.

I acknowledge that I have read the rebuttal(s).

**Questions:**

In this paper, the authors proposed a new method, called LFDe, for EE. To expedite the pre-training phase, the authors proposed to automatically construct a small amount of weak-label data that is used during the pre-training to quickly familiarize with the task format. This small pre-training data is used to train a prompt-based sequence labeling model (P-SLM) that is introduced for both event detection and argument extraction phases using type-aware prompts and role-aware prompts. P-SLM is further fine-tuned on the target dataset. The proposed method was shown to be effective and efficient mainly for low-resources cases. There are some points that should be taken into consideration:

1. GDELT is used for pre-training. The choice of this particular data should be justified. Given the importance of this dataset in the pre-training phase, several statistics regarding this dataset should be clearly reported and discussed in the paper, such as the event types, triggers, arguments, roles, etc, covered by this dataset. In addition, how can we make sure that there is no data leakage from the testing set in this dataset? Given the huge improvement obtained from pre-training, we need to clearly understand the pre-training data. CLEVE [1] uses different pre-training data. How different are the results when using CLEVE’s pre-training data? If the choice of pre-training data is important, we need to understand how to make such a choice.

2. The event schema which is integrated as prompts into the input is an important part of the model, but it is not clear how this event schema is constructed. Is the event schema constructed manually or automatically? Is GDELT used to construct the event schema? How can we make sure that there is no data leakage from the testing set in the event schema and by consequence in the constructed prompts? How many examples in the type-aware prompts are needed?

3. For the fully fine-tuning case (Table 5), it is not clear why the event schema is not as helpful as in the low-resource scenarios. If more trigger examples and role examples are included in the type-aware prompts, do we expect an increase in the evaluation metrics?

[1] Wang et al. CLEVE: Contrastive Pre-training for Event Extraction

**Reviewer Confidence:**

3: The reviewer is confident but not certain that the evaluation is correct

**Scope:**

3: The work is somewhat relevant to the Web and to the track, and is of narrow interest to a sub-community

---

### Official Review · Reviewer_mpwh · 2023-11-20

**Novelty:** 5
**Technical Quality:** 5

**Review:**

The paper addresses the computational challenges of pre-training Event Extraction (EE) models by introducing an innovative framework that significantly reduces the reliance on large-scale labeled data and computational resources. The main contributions of this paper are the development of a novel, task-oriented pre-training approach that employs automatically generated weak-label data to quickly familiarize the EE models with the task at hand, and the implementation of a prompt-based sequence labeling model that efficiently identifies events within texts by seeking semantic congruence with type-aware prompts. This model refines the conventional sequence labeling process, rendering it more suitable for scenarios with limited data. The paper showcases that this method not only diminishes the requirement for pre-training instances, training duration, and memory consumption by notable margins but also enhances the performance of EE tasks across different levels of data scarcity, as evidenced by the experiments conducted on real-world datasets.

**Pros:**

1. The paper proposes a novel pre-training framework that mitigates the cost and complexity associated with event extraction tasks by reducing the dependency on annotated data.

2. It introduces a relatively new prompt-based sequence labeling model for event extraction, redefining the task to focus on identifying words or phrases semantically closest to the prompt within a given sentence.

3. The paper includes a comprehensive experimental analysis, comparing a rich set of baseline methods and demonstrating the advantages of the proposed framework in low-resource settings, thereby affirming its effectiveness in scenarios with scarce data.


**Cons:**

1. The method proposed in the article may appear as an incremental work that combines existing mature technologies rather than presenting groundbreaking innovation. Especially under fully supervised conditions, the method does not outperform other works mentioned in the manuscript.

2. Despite claims of requiring fewer samples and faster training times during the pre-training phase, such advantages may not be directly comparable to methods like UIE, which include data from multiple tasks such as NER and RE, inevitably resulting in more extensive training datasets and longer training times. The direct comparison lacks fairness because these methods demonstrate capabilities in NER and RE tasks that the model presented in the paper does not have.

3. While the prompt-based sequence labeling may enhance model performance, generating specific prompts for each event type and argument role and then performing inference to obtain results undoubtedly adds to the computational burden during inference. The paper does not clarify the extent of this inference overhead nor how it compares to other methods. Moreover, in practical applications, inference latency is often more critical than training costs.

**Questions:**

See cons.

**Reviewer Confidence:**

3: The reviewer is confident but not certain that the evaluation is correct

**Scope:**

3: The work is somewhat relevant to the Web and to the track, and is of narrow interest to a sub-community

---

### Official Review · Reviewer_4cNb · 2023-11-21

**Novelty:** 6
**Technical Quality:** 6

**Review:**

The paper presents a novel approach named LFDe (Lighter, Faster, and more Data-efficient pre-training framework) for the Event Extraction (EE) task. This approach is designed to address the challenges of existing pre-training methodologies for EE, which typically require significant computational resources, high-performance hardware, and extensive training duration. LFDe distinguishes itself by deviating from the convention of establishing a comprehensive representation space during pre-training. Instead, it focuses on rapidly acquainting with the task format using a small quantity of automatically constructed weak-label data, where the strategy is aimed at reducing the reliance on costly and labor-intensive data annotation. By introducing this framework, the paper proposes a solution that is faster and lighter in terms of computational demand and also more data-efficient, compared to traditional methods in the field of EE.

-------------


## Strength:

- **Innovative Use of Weak-Label Data**: LFDe's use of automatically constructed weak-label data significantly reduces the dependency on large, manually annotated datasets. This approach incorporates Part-of-Speech (POS) tagging and Abstract Meaning Representation (AMR) to effectively identify pseudo-triggers and arguments in unlabeled texts. By leveraging such weak-label data, LFDe offers a more efficient and scalable way to train models for Event Extraction (EE), particularly in scenarios where annotated data is scarce.

- **Novel Model Framing for EE**: The paper introduces a novel model that redefines EE as identifying words or phrases closest in meaning to a given prompt within a sentence. This new framing allows the model to adapt quickly to EE tasks without the need for extensive representation space modeling. This adaptability enhances the model's applicability across diverse scenarios, potentially making it a versatile tool for EE.

- **Efficiency in Training and Resource Usage**: LFDe is demonstrated to require significantly fewer pre-training instances and a shorter pre-training period, all while utilizing less memory. Also it achieves this without sacrificing performance, even in data scarcity scenarios. This efficiency makes LFDe a practical and resource-effective solution for EE, particularly valuable in contexts where computational resources are limited.

-------------

## Weakness:
- **Pipeline Complexity and Error Propagation**: While the pipeline of the proposed framework is logically structured, it involves multiple steps such as POS tagging, WordNet filtering, etymology dictionary requests, AMR parsing, and relying on Large Language Models (LLMs). And it thus increases the complexity of the method. This complexity not only increases the likelihood of errors in each step but also raises concerns regarding the cost and accessibility of external resources, particularly the expensive APIs of tools like ChatGPT. Addressing these concerns is crucial for the practical deployment of LFDe.

- **Lack of Quantitative Efficiency Metrics**: To convincingly demonstrate the efficiency of LFDe, quantitative metrics such as GFLOPs and the number of parameters should be provided. These metrics are essential to reflect the cost of computational resources, offering a more comprehensive understanding of the model's efficiency.

- **Details on Rule-Based Pseudo Data Generation**: The effectiveness of LFDe hinges on the quality of pseudo data generation, which is primarily rule-based. However, the paper introduces the rule-based pseudo-triggers annotation (as in Section 4.3.1) only briefly and quite roughly. A comprehensive list of rules for non-notional verb filtering, discarding nouns ending in specific suffixes, and a comparison of AMR-based pseudo roles/arguments with real roles/arguments in evaluation datasets (like ACE05, ERE) would be invaluable. Such detailed information would help readers gauge the similarity between the two sets and appreciate the model's data generation strategy.

- **Pre-Training Data Specifications**: Section 5.1.1 mentions roughly 30,000 weak-label instances used for pre-training. More detailed information, such as the number of sentences, average sentence length, and the number of events/arguments per sample, would provide a clearer picture of the training dataset. This level of detail is important for assessing the representativeness and diversity of the training data.

- **Clarification in Equation**: In Equation 2, there seems to be an inconsistency in the notation used for denoting sentences in set $S$. If $s^{ti}_j$ denotes $s_1$ in S, as mentioned in line 463, it would be beneficial to maintain consistency in notation throughout the paper to avoid confusion.

-------------
**Increased rating after author response**:
I am quite satisfied with the merits this work and its authors bring to me and have no further concerns, so I have decided to increase the rating further.

**Questions:**

- See Weakness.
- +Question: In Equation 2, does the notation $s^{ti}_j$ denote $s_1$ in $S$? If yes, it should keep consistent with the ones in line 463.

**Reviewer Confidence:**

4: The reviewer is certain that the evaluation is correct and very familiar with the relevant literature

**Scope:**

4: The work is relevant to the Web and to the track, and is of broad interest to the community

---

### Official Review · Reviewer_WmaN · 2023-11-23

**Novelty:** 4
**Technical Quality:** 3

**Review:**

This paper proposes a new pre-training framework for the EE task, named LFDe.

The writing is cumbersome, hard to follow, full of adjectives praising the own solution. The contributions are hard to grasp.

The method development is difficult to flow but seems correct.

The experimental evaluation has serious issues. It seems that the authors run all experiments in a  single split of the dataset without repetitions. This means that there is a single validation and  test sets and assessing the  generalization of the methods with different partitions of the datasets is not possible,
´
There is also no statistical treatment of the results; no statistical significance tests. Without those it is impossible to rule out the null hypothesis of equality of results, especially in face of very close results when compared to the baselines.

**Questions:**

Please check [1] to understand the importance of the right experimental protocols-
[1] Washington Cunha, Vítor Mangaravite, Christian Gomes, Sérgio D. Canuto, Elaine Resende, Cecilia Nascimento, Felipe Viegas, Celso França, Wellington Santos Martins, Jussara M. Almeida, Thierson Rosa, Leonardo Rocha, Marcos André Gonçalves:
On the cost-effectiveness of neural and non-neural approaches and representations for text classification: A comprehensive comparative study. Inf. Process. Manag. 58(3): 102481 (2021)

**Reviewer Confidence:**

3: The reviewer is confident but not certain that the evaluation is correct

**Scope:**

3: The work is somewhat relevant to the Web and to the track, and is of narrow interest to a sub-community

---

### Decision · Program_Chairs · 2024-01-22

**Decision:**

Accept

**Comment:**

This paper proposes a new lighter, faster and data-efficient pre-training framework for the event-extraction task.

 Pros:
 - Reduction of dependency on large, manually annotated datasets through use of weakly-labeled data
 - Novel event extraction model
 - Reduction in pre-training instances and shorter pre-training period, with reduced memory

 Cons:
 - Concerns about experimental evaluation and a lack of statistical significance
 - Concerns about pipeline complexity
 - Unclear quantitative computational cost measures
 - Additional data from rule-based pseudo-data generation would be valuable.
 - Some detail missing from training dataset reflecting representativeness and diversity.

 Overall, the authors have provided useful responses and multiple reviewers were positive from the start. While there are areas for which this paper would benefit from additional attention, it already has sufficient value to be accepted here.